# Tumor-Infiltrating Lymphocytes in Resected Esophageal and Gastric Adenocarcinomas Do Not Correlate with Tumor Regression Score After Neoadjuvant Chemotherapy: Results of a Case-Series Study

**DOI:** 10.3390/cancers16213694

**Published:** 2024-11-01

**Authors:** Fotios Seretis, Chrysoula Glava, Spyridon Smparounis, Dimitra Riga, Georgios Karantzikos, Maria Theochari, Dimitrios Theodorou, Tania Triantafyllou

**Affiliations:** 11st Propaedeutic Department of Surgery, Hippokrateion General Hospital of Athens, National and Kapodistrian University of Athens, 11527 Athens, Greece; spyros.smparounis@gmail.com (S.S.); t_triantafilou@yahoo.com (T.T.); 2Department of Pathology, Hippokrateion General Hospital of Athens, 11527 Athens, Greece; chryssa.g@hotmail.com (C.G.); dimitra1402@hotmail.com (D.R.); 3Department of Surgery, Hippokrateion General Hospital of Athens, 11527 Athens, Greece; karantzikos@hotmail.com; 4Department of Medical Oncology, Hippokrateion General Hospital of Athens, 11527 Athens, Greece; mtheochari@gmail.com

**Keywords:** tumor infiltrating lymphocytes, tumor regression, esophageal, gastric, cancer, chemotherapy, neoadjuvant

## Abstract

Adenocarcinomas of the esophagogastric junction and stomach pose significant morbidity and mortality burdens for patients worldwide. Neoadjuvant treatments are increasingly being utilized before the performance of radical surgeries, with varying degrees of response. We attempted to describe how lymph nodes regress after neoadjuvant chemotherapy in comparison to primary tumors. We performed an analysis of tumor-infiltrating lymphocytes on pathology slides of esophagectomy and gastrectomy patients from a single-institution cohort. Our research suggests that tumor-infiltrating lymphocytes correlate differently with the degree of response of lymph nodes versus the degree of response of primary tumors, suggesting that different biologic processes are involved. We investigated the impact of neoadjuvant chemotherapy, namely the FLOT regimen, on lymph node regression and its correlation with tumor-infiltrating lymphocytes.

## 1. Introduction

Both esophageal and gastric cancers represent entities with complex decision-making entailed in their management, requiring a multidisciplinary approach. The CROSS trial established preoperative chemoradiotherapy for esophageal or junctional cancer [1], while perioperative chemotherapy with FLOT4 (fluorouracil, leucovorin, oxaliplatin, and docetaxel) was also widely introduced and accepted in the medical community for the management of locally advanced, resectable gastric or gastroesophageal junction adenocarcinomas [2]. While the CROSS regimen was associated with increased numbers of pathologic complete response compared to FLOT [3], this difference cannot be translated into increased overall survival at long follow-up. In another study, both regimens appeared to be associated with comparable long-term survival outcomes [4] for distal esophageal or esophagogastric junction adenocarcinomas, although the CROSS regimen appeared to be associated with higher rates of distal nodal recurrence. Tailored-made decision-making is further supported by results from the European multicenter ENSURE trial, according to which neoadjuvant chemoradiotherapy, compared to neoadjuvant chemotherapy, correlates with a reduced probability of local recurrence but reduced distant recurrence-free survival, despite no differences in the overall survival [5]. The advent of immunotherapy in the treatment algorithms appears to complicate matters for the worse. The CheckMate 649 trial investigated first-line checkpoint inhibitor nivolumab (PD-1 inhibitor) added to chemotherapy in advanced gastric, gastroesophageal junction, and esophageal adenocarcinomas [6]. Recently published results from this trial at 3 years follow-up support the addition of immunotherapy to chemotherapy in these groups of patients, as it is associated with increased overall survival. In the perioperative setting, however, neoadjuvant and adjuvant pembrolizumab (PD-1 inhibitor) do not appear to confer a better survival, despite being associated with increased pathologic complete response rates [7], according to recently published results from the KEYNOTE-585 trial. The ATTRACTION-5 trial also failed to demonstrate any benefit from adjuvant immunotherapy plus chemotherapy versus adjuvant chemotherapy alone after gastrectomy with D2 or more lymph node dissection in gastric or gastroesophageal junction adenocarcinomas [8]. In this phase 3 trial, adjuvant nivolumab (an PD-L1 inhibitor) administration in conjunction with S1 (tegafur/gimeracil/oteracil) or CapeOx (capecitabine-oxaliplatin) failed to meet the trial primary endpoint set for relapse-free survival.

Tumor response to neoadjuvant treatment represents a major determinant of prognosis. The pathologic response after neoadjuvant treatment in patients with esophageal cancer [9] appears to predict long-term survival. More specifically, a complete pathologic response has been associated with improved survival in patients with esophageal cancer [10] and for gastric adenocarcinoma [11]. On the contrary, non-response to neoadjuvant chemoradiation for esophageal cancer does not benefit from neoadjuvant treatment at all, compared to patients undergoing upfront surgery [12]. A recent systematic review reported that up to one-third of patients with esophageal cancer receiving neoadjuvant chemo (radiation) achieved a pathologic complete response [13]. The same appears to hold true for gastric or gastroesophageal junction adenocarcinomas receiving neoadjuvant chemotherapy [14].

The tumor microenvironment has been studied to further understand the biologic behavior of tumors. More specifically, tumor-infiltrating lymphocytes have been implicated to play a role in gastric cancer [15], with increasing tumor T and N stages being correlated with decreased tumor-infiltrating lymphocytes (TILs). TILs have also been associated with Helicobacter pylori infection and mismatch repair status [16]. TILs have also been shown to be of prognostic value in patients undergoing esophagectomy following neoadjuvant DCF (docetaxel, cisplatin, and 5-fluorouracil) [17].

When focusing on esophageal adenocarcinomas and gastric adenocarcinomas, there are fewer existing data. A study published by Schoemmel et al. [18] on locally advanced esophageal adenocarcinoma patients analyzed the spatial distribution of TILs. More specifically, the authors described that a rich T-cell infiltrate around the tumor center is associated with a better survival, while TILs at the invasion zone do not seem to be correlated with survival. Pre-treatment TILs appear to also influence the response to neoadjuvant chemoradiotherapy in esophageal adenocarcinoma [19], and increasing TILs appear to correlate with better tumor regression scores and pathologic complete responses. With regards to gastric adenocarcinoma, TILs have been found at variant degrees across the four molecularly classified subtypes of gastric cancers (namely, Epstein–Barr virus positive, mismatch repair-deficient, aberrant TP53, and other) [20].

We conducted a study on patients with gastric and esophageal adenocarcinomas to further investigate whether there is a correlation between tumor regression scores and TILs.

## 2. Materials and Methods

Records from patients with gastric or esophageal adenocarcinoma operated on in a single academic institution during the time period from January 2022 until January 2024 were reviewed after institutional review board permission. Demographics, including age and gender, were recorded for all patients. Only patients with adenocarcinomas who underwent esophagectomy or gastrectomy according to respective tumor location were included. Only patients who were operated on after the completion of the neoadjuvant FLOT regimen were included in the analysis. Type of operation performed and pathology reports/relevant pathology slides were reviewed, including total number of lymph nodes, positivity or negativity of lymph nodes, node positivity after regression, and node negativity with regression. Tumor regression (TRG) score was reported according to the Mandard classification [21]. TRG values ranged from 1–5, with 1 designating no residual cancer cells, TRG 2 residual cancer cells, TRG 3 fibrosis outgrowing residual cancer, TRG 4 residual cancer outgrowing fibrosis, and TRG 5 absence of regressive features. Staging was performed according to the American Joint Commission on Cancer (AJCC), 8th edition [22]. All relevant slides were also reviewed to calculate TIL scores for all patients, expressed as a numerical score, according to relevant guidelines from international consortia on tumor-infiltrating lymphocytes [23]. When no tumor could be identified after neoadjuvant treatment, by definition, no TIL scores could be assigned. More specifically, a review of the pathology slides measurement of TILs did not require any stains on the slides but rather only calculating TIL scores from “measuring” the number of lymphocytes relevant to the tumor. The reader is directed to the aforementioned reference for the exact protocol used to calculate TIL scores. Pathology reports for all relevant operations were reviewed, and pathology slides were re-reviewed by 2 independent expert pathologists (CG and DR) to calculate the TIL score. A TIL score from 0–80 was granted, and where no tumor was found, “no tumor” was designated for TIL score calculations. Figure 1 and Figure 2 depict slides from our patient cohort, with the first corresponding to a TIL-high patient, while the latter originates from a TIL-low patient.

The SPSS package [IBM SPSS statistics version 29.0.2.0(20)] was used for all statistical analyses. In order to create subgroups of TIL patients of sufficient sample size, we created two groups for TILs, namely TIL-low and TIL-high groups, with the TIL-low group containing scores from 0–30 and the TIL-high group containing numbers from 40–80. Representative pictures from pathologic slides from our patient series are provided in pictures 1 (TIL-low) and 2 (TIL-high). We affirmed that both TIL-low and TIL-high groups contained more than 25 patients each, and therefore, by definition, we assumed that the sample size followed a normal distribution and included categorical data; thus, Spearman’s r correlative statistic could be used for analysis. With regards to the Mandard score, we stratified patients into a “good response” TRG1 and TRG2 and a “bad response” TRG 3, TRG 4, and TRG 5. Correlation analysis between TRG and the TIL-low or TIL-high group was performed using the Spearman’s r statistic and Kendall’s tau statistics.

Data on lymph node counts were extracted for all patients, including the total number of positive lymph nodes and total number of negative lymph nodes. More importantly, we sought to identify the number of nodes that were “turned into” negative after neoadjuvant chemotherapy, namely negative lymph nodes with regression characteristics. Nodes continuing to be positive with regression characteristics were also identified.

## 3. Results

A total of 76 patients with adenocarcinomas of the esophagogastric junction or the stomach who underwent neoadjuvant chemotherapy were identified from a retrospective review of a prospectively maintained institutional database. Data on the operative note from one patient were missing, but the pathology slides were available, so we chose to include the patient as “missing data” on the relevant tables regarding the type of operation (esophagectomy versus gastrectomy). Complete results were available on 75 patients. We chose to include the patient with the missing operative data because the data regarding histology were available, and thus, tumor and lymph node regression scores could be derived, as well as TIL scores. We focused on the adenocarcinoma esophagogastric junction/stomach histology as a single entity, and thus, we felt that the patient with lacking operative notes data could be included for analysis. Most of the patients had male gender assigned at birth (64%), with the rest being female. A total of 51 patients suffered from esophagogastric junction adenocarcinomas and the other 23 from gastric ones. Patients underwent either gastrectomy or esophagectomy as part of multi-disciplinary management. All patients received four cycles of FLOT (fluorouracil, leucovorin, oxaliplatin, and docetaxel) chemotherapy preoperatively. Data on the type of operation for one patient were missing, but the patient was included in the analysis. The relevant percentages regarding age, tumor location, and operation performed, respectively, are demonstrated in the tables. A total of 66% of patients underwent esophagectomy and the rest gastrectomy; data were missing for one patient.

Results of Mandard scores for all patients are shown in Table 1. Results regarding the TIL scores for all patients are shown in Table 2. TIL scores ranged from 0 to 80 in our case series. In eleven (11) patients, no tumor was identified on pathology slides; therefore, the TIL score was designated as “no tumor” for them. When classifying patients as being in the TIL-low or TIL-high group, 41 patients were identified in the low group versus 17 in the high group (Table 3). Correlation analysis between TRG and the TIL-low or TIL-high group was performed using the Spearman’s r statistic and Kendall’s tau statistics. In both metrics, no statistically significant correlation could be established. Levene’s test was used to assess the equality of variances for the TRG variable calculated for TIL-low and TIL-high groups (Table 4). The *p*-value was calculated to be above 0.05; thus, no difference between the group variances could be assumed. To test whether the failure to detect a statistically significant correlation was due to small sample size, we performed a power analysis calculation (Table 5). Our power analysis results confirmed that failure to demonstrate statistically significant correlation between the TIL-high/TIL-low group and TRG was not due to sample size.

Data on lymph node counts were extracted for all patients, including total number of positive lymph nodes and total number of negative lymph nodes. Results were also extracted for lymph nodes that showed regression characteristics and were classified as positive or negative. Results are depicted in Table 6. The mean number of lymph nodes examined was 29, which is in accordance with international standards for lymph node yield after curative intent surgery for adenocarcinoma of the esophagogastric junction or adenocarcinoma of the stomach. No correlation was identified between lymph node positivity or negativity and tumor-infiltrating lymphocytes, with the relevant *p*-values being >0.05 using Pearson’s correlation statistic. However, a statistically significant positive correlation was identified between tumor-infiltrating lymphocytes and both positive nodes with regression characteristics and negative nodes with regression characteristics (*p*-value < 0.05 for the two-sided test). Relevant results are shown in Table 7, Table 8 and Table 9.

## 4. Discussion

Tumor-infiltrating lymphocytes present a component of the tumor microenvironment, affected not only by the tumor per se, but also by the host immune reaction to cancer cells. It seems, therefore, prudent to explore their role in gastric and esophagogastric junction adenocarcinomas. In gastric adenocarcinoma patients, lymphocytic infiltration appears to be inversely correlated with peri-tumoral and intra-tumoral budding, as well as a more immature desmoplastic reaction [24]. In the same report, tumor budding appears to correlate with worse survival outcomes. If one considers the landmark publication from Hanahan and Weinberg on the hallmarks of cancer [25], the findings in the aforementioned study might be explained with a teleologic point of view. It might be logical to assume that in patients in whom clusters of tumor cells (tumor buds) evaded the immune system, facilitated by the acquirement of necessary biologic properties and/or by their crosstalk with a tumor-promoting desmoplastic microenvironment, a poor lymphocytic infiltrate merely represents the hallmark of an overwhelmed immune system by the tumor, thus explaining why these patients fare worse in terms of survival outcomes. Spatial distribution of the tumor lymphocytic infiltrate has been studied in esophageal carcinomas, suggesting that the disease state is strongly affected by the tumoral and para-tumoral immune signatures and that this spatial distribution is correlated with lymph node stage, tumor stage, and Mandard score [26]. T-cell exhaustion and/or blockade through various tumor or tumor-environment mechanisms have been described in esophageal cancer patients [27]. Tumor microenvironment and cancer-associated fibroblasts also appear to influence the type and magnitude of host immune system and tumoral interactions [28]. To complicate matters more, immunotherapy with immune checkpoint inhibitors appears to be efficacious for gastric cancer and gastroesophageal junction cancer in the setting of advanced disease [29]. Moreover, a recent study also reported on favorable long-term outcomes with immune check-point inhibitors for deficient MMR (mismatch repair) gastric cancers [30]. A recent report including patients from the National Cancer Database suggested a benefit for adjuvant immunotherapy for patients with esophageal adenocarcinomas undergoing neoadjuvant chemoradiotherapy followed by surgery, compared to no immunotherapy or neoadjuvant immunotherapy [31]. Results from this large cohort of patients might imply that “releasing the brakes” on tumor-associated immune response might be favorable after tumor burden depletion with neoadjuvant treatment and surgery.

We tried to elucidate the role of tumor-infiltrating lymphocytes in patients with adenocarcinomas of the gastroesophageal junction or stomach after neoadjuvant chemotherapy. We did not identify any correlation between tumor-infiltrating lymphocytes and tumor regression scores, suggesting that the two are independent processes. Nodal status in our cohort also did not correlate with TILs. Most interestingly, tumor-infiltrating lymphocytes correlated with lymph node positivity and with regression features, as well as with lymph nodes regressing into negative status. It appears that the lymph node response to chemotherapy, whether complete (lymph node negativity with regression characteristics) or incomplete (lymph node positivity with regression characteristics), provokes an infiltrate of T cells. We therefore demonstrated a differential role for TILs with regards to primary tumor and tumor-draining lymph nodes. It might, therefore, seem prudent to consider the response to neoadjuvant treatment in lymph nodes as a distinct process from primary tumor response. A recent multicenter study in the United Kingdom on esophageal adenocarcinomas reported that pathologic lymph node regression after neoadjuvant chemotherapy predicts recurrence and survival beyond classic TNM (tumor node metastasis) staging and primary tumor regression grading [32]. Authors from another study on esophageal adenocarcinoma patients also concluded that lymph node regression is a strong prognostic factor and that it may be even more important than primary tumor response [33]. Reim et al. [34] reported on 480 gastric and esophagogastric junction adenocarcinomas and found that lymph node regression after neoadjuvant treatment correlates with improved prognosis. More specifically, in that study, patients with lymph node negativity with regression characteristics had better prognoses than patients with lymph node positivity with regression characteristics. Our results might explain the findings reported in these studies. The improved prognosis with lymph node regression might be at least partially explained by the associated tumor-infiltrating lymphocytes it provokes.

Unfortunately, our study also suffers from limitations. A retrospective review of single-institution data carries an inherent selection bias risk. Another limitation is the lack of follow-up data for the survival data. Also important is the lack of data on the MSI status of patients, despite its relatively low incidence in gastroesophageal junction and gastric adenocarcinomas [35,36]. Our study, however, provides unique data on patients with adenocarcinoma histology only and provides detailed information, not only on tumor-infiltrating lymphocytes in relation to tumors but also in relation to lymph nodes after the same neoadjuvant chemotherapy regimen for all patients.

## 5. Conclusions

Tumor-infiltrating lymphocytes appear to have no correlation with tumor regression after neoadjuvant chemotherapy in esophagogastric junction and gastric adenocarcinoma patients, suggesting that the two processes are distinct and might be based on different biology. We demonstrated that TILs correlate with lymph node regression, whether complete or partial, suggesting that lymph node response provokes an inflammatory response of the host after neoadjuvant chemotherapy. Our study suffers from limitations due to its small size and retrospective nature but presents unique data in a field where they are scarce. It has important implications for cancer research, especially tumor–host immune system interactions.

## Figures and Tables

**Figure 1 cancers-16-03694-f001:**
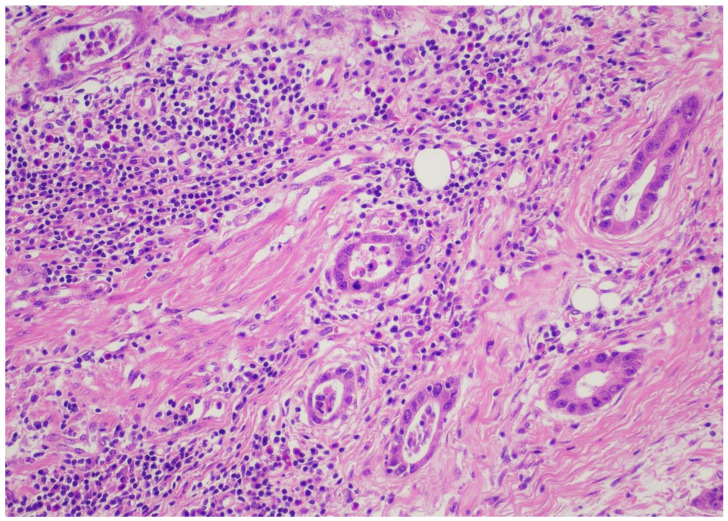
TIL-high patient.

**Figure 2 cancers-16-03694-f002:**
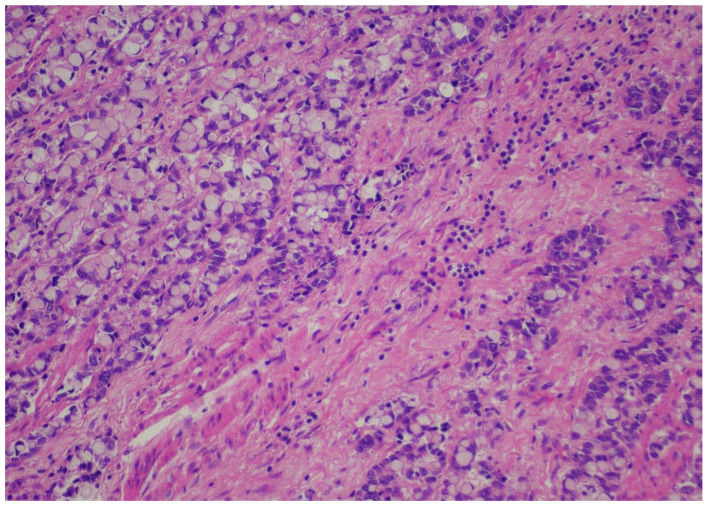
TIL-low patient.

**Table 1 cancers-16-03694-t001:** Patient characteristics.

Characteristics	Number	Frequency (%)
**Gender**		
F M	11 64	14.7 85.3
**Location**		
“Missing” EGJ gastric	1 51 23	
**Operation**		
“Missing data”	1	1.3
esophagectomy	50	66.7
gastrectomy	24	32.0
**TRG Mandard**		
1 2 3 4 5	11 14 19 21 9	14.7 18.7 25.3 28.0 12.0
**TILs**		
TIL-low TIL-high No tumor	40 17 11	
**TOTAL**	75	

**Table 2 cancers-16-03694-t002:** Tumor-infiltrating lymphocyte score demographics.

		Frequency	Percent
Valid	Missing data	6	8.0
	0	8	10.7
	5	8	10.7
	10	15	20.0
	20	6	8.0
	30	4	5.3
	40	6	8.0
	50	2	2.7
	55	1	1.3
	60	1	1.3
	70	2	2.7
	80	5	6.7
	no tumor	11	14.7
	Total	75	100.0

**Table 3 cancers-16-03694-t003:** Percentages of TIL-low and TIL-high groups in TRG groups.

Case Processing Summary							
	TIL_GROUP	Cases					
		Valid		Missing		Total	
		N	Percent	N	Percent	N	Percent
TRG_Group	TIL Low	40	97.6%	1	2.4%	41	100.0%
	TIL High	17	100.0%	0	0.0%	17	100.0%

**Table 4 cancers-16-03694-t004:** Correlation between TRG and TIL expressed in Kendall’s tau and Spearman’s r statistics.

Correlations				
			TRG_Group	TIL_GROUP
Kendall’s tau_b	TRG_Group	Correlation Coefficient	1.000	−0.070
		Sig. (2-tailed)	0.0	0.601
		N	74	57
	TIL_GROUP	Correlation Coefficient	−0.070	1.000
		Sig. (2-tailed)	0.601	0.0
		N	57	58
Spearman’s rho	TRG_Group	Correlation Coefficient	1.000	−0.070
		Sig. (2-tailed)	0.0	0.605
		N	74	57
	TIL_GROUP	Correlation Coefficient	−0.070	1.000
		Sig. (2-tailed)	0.605	0.0
		N	57	58

**Table 5 cancers-16-03694-t005:** Power calculation analysis for the TIL-–TRG correlation analysis.

Power Analysis Table					
	Power ^b^	Test Assumptions			
		N	Null	Alternative	Sig.
Spearman Correlation ^a^	0.968	57	0	0.5	0.05

a. Two-sided test. b. Based on Fisher’s z-transformation and normal approximation. The variance estimation is based on the method suggested by Bonett and Wright.

**Table 6 cancers-16-03694-t006:** Demographics of lymph nodes from surgical specimens examined.

Number of Positive Nodes (Total)	Number of Patients	Percent of Patients
0	44	58.7
1	9	12.0
2	5	6.7
3	3	4.0
4	1	1.3
5	4	5.3
6	1	1.3
7	1	1.3
8	2	2.7
11	1	1.3
14	1	1.3
15	1	1.3
19	1	1.3
47	1	1.3
**Number of positive nodes with regression characteristics**		
0	54	72.0
1	12	16.0
2	3	4.0
4	2	2.7
5	2	2.7
6	1	1.3
13	1	1.3
**Number of negative lymph nodes with regression characteristics**		
0	45	60.0
1	9	12.0
2	4	5.3
3	3	4.0
4	3	4.0
5	3	4.0
6	4	5.3
7	1	1.3
8	1	1.3
14	1	1.3
**Lymph nodes in the surgical specimen**	**Mean**	**Std. Deviation**
10–59	29.24	10.151

**Table 7 cancers-16-03694-t007:** TIL correlations with positive lymph nodes.

Correlations			
		TIL_GROUP	total positive nodes
TIL_GROUP	Pearson Correlation	1	−0.090
	Sig. (2-tailed)		0.503
	N	58	58
total positive nodes	Pearson Correlation	−0.090	1
	Sig. (2-tailed)	0.503	
	N	58	75

**Table 8 cancers-16-03694-t008:** TIL correlations with negative lymph nodes.

Correlations			
		TIL_GROUP	total negative nodes
TIL_GROUP	Pearson Correlation	1	0.219
	Sig. (2-tailed)		0.099
	N	58	58
total negative nodes	Pearson Correlation	0.219	1
	Sig. (2-tailed)	0.099	
	N	58	75

**Table 9 cancers-16-03694-t009:** TIL correlations with positive lymph nodes with regression characteristics and with negative lymph nodes with regression characteristics.

Correlations				
		TIL_GROUP	positive nodes with regression characteristics	negative nodes with regression characteristics
TIL_GROUP	Pearson Correlation	1	0.159	0.322
	Sig. (2-tailed)		0.233	0.014 *
	N	58	58	58
positive nodes with regression characteristics	Pearson Correlation	0.159	1	0.380
	Sig. (2-tailed)	0.233		<0.001 **
	N	58	75	74
negative nodes with regression characteristics	Pearson Correlation	0.322	0.380	1
	Sig. (2-tailed)	0.014 *	<0.001 **	
	N	58	74	74

* Correlation was significant at the 0.05 level (2-tailed). ** Correlation was significant at the 0.01 level (2-tailed).

## Data Availability

Data are not publicly available due to constraints to ensure patient confidentiality.

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
