# Peer review of "Tumor-Infiltrating Lymphocytes in Resected Esophageal and Gastric Adenocarcinomas Do Not Correlate with Tumor Regression Score After Neoadjuvant Chemotherapy: Results of a Case-Series Study"

_cancers, 2024, doi:10.3390/cancers16213694_

Round 1
Reviewer 1 Report
Comments and Suggestions for Authors
Dear Authors
Manuscript describes well about “Tumor infiltrating lymphocytes in resected esophageal and gastric adenocarcinomas do not correlate with tumor regression score after neoadjuvant chemotherapy - 75 patients with gastric and esophageal adenocarcinomas to further investigate whether there is a correlation between tumor regression scores and tumor-infiltrating lymphocytes (TILs).
The following steps should more clear information for readers to enjoy it,
1) Please keep the Abstract section.
2) Institutional Review Board approval code number is missing.
3) Statistics section missing in the materials and methods.
4) 75 patients with adenocarcinomas of the esophagogastric junction or the stomach who had undergone neoadjuvant chemotherapy were identified from retrospective review of a prospectively maintained institutional database – Please mention year in the manuscript.
Comments on the Quality of English LanguageModerate editing of English language required.
Author Response
Thank you for your comments. We have undertaken all major revisions according to your suggestions
1)Abstract session has been added
2) Institutional review board has been added
3) Statistics session added in materials and methods
4) Year of review added
Reviewer 2 Report
Comments and Suggestions for Authors
Seretis et al in their research show that tumor-infiltrating lymphocytes (TILs) in resected esophageal and gastric adenocarcinomas do not correlate with tumor regression scores after neoadjuvant chemotherapy. Nevertheless, they show a clear correlation between TILs and positive/negative nodes with regression characteristics. Research on TILs in resected esophageal and gastric adenocarcinomas, particularly in the context of neoadjuvant chemotherapy, is of significant importance because it might have an effect on long-term survival. However, the prognostic role of TILs in EC and GC remains controversial, varying with the distribution site and cell types. In addition, an increase in TILs after chemotherapy is particularly important because it suggests that the chemotherapy not only directly targets the tumor cells but also enhances the body's immune response against the tumor that can not only be seen through tumor regression scores but also through nodes with regression characteristics.
With that being said, the manuscript needs extensive revision mainly in the results section. Because all the important data is lost in unnecessary tables.
Major concerns:
Simple summary and Abstract are missing. This should be corrected.
All statements of contribution, funding, conflict of interest, data sharing etc. are also missing
Introduction section:
1) Line 39: Does ATTRACTION-5 also refers to targeting the PD-1/PD-L1 pathway or some other such as CTLA-4? Is there some data on other immune checkpoint inhibitors?
Material and methods section lacks:
1) Ethical approval statement (was written consent obtained from patients?), ethical board approval ( coming from the hospital?), and approval number are all missing
2) Description of how “all relevant slides were reviewed again” and how were scores and TILs calculated? How were TILs visualized? Classical H&E staining and size comparison or some other immunohistochemical markers (eg. CD45, CD3, CD4, CD8, CD19, CD16, CD56?)
3) The statistical analysis description should be added to M&M section. It is partially described in Line 128: „SPSS package [IBM SPSS statistics version 29.0.2.0(20)] was used for all statistical analysis” and then later throughout the result (software, statistical tests, p-values). However, it should be moved to M&M to make results interpretation easier.
Results section:
1) Lines 91-97 should go into M&M. And results should start with the description of clinicopathological characteristics.
2) Why are you including in the research/results one patient where information on whether he had esophagogastric junction adenocarcinomas or gastric one is missing?
3) Is it necessary to include cumulative percent? What does it add to the results value?
4) Clinicopathological features of patients included in the study should all go into one table. I suggest to omit table 5 and just add into one table TILs low and TILS high groups. Because even though you have the data you are obviously not using it for any further analysis
e.g.:
|
Characteristics |
N (%) |
|
Gender |
|
|
Male |
X (%) |
|
Female |
X (%) |
|
Tumor operation |
|
|
esophagectomy |
X (%) |
|
gastrectomy |
X (%) |
|
Tumor location |
|
|
EGJ |
X (%) |
|
Gastric |
X (%) |
|
TRG Mandard |
|
|
1 |
X (%) |
|
2 |
X (%) |
|
3 4 |
X (%) X (%) |
|
5 |
X (%) |
|
TILs |
|
|
low |
X (%) |
|
high |
X (%) |
Etc
5) The same goes for the Correlations tables. Their importance is completely lost between Case Processing Summary, Test of Homogeneity of Variance, Confidence Intervals of Kendall's tau_b, Confidence Intervals of Spearman's rho, Power Analysis Table, … Which are all important data on statistics but should be omitted/mentioned that they were done or added as supplementary data.
6) In Table 14, asterisks “*” and “**” are added in the wrong row, they should go where the significance is- next to .014 and <.001.
7) Based on what was TIL-low and TIL-high group defined? Can you describe it in the M&M section?
8) Can you add some representative microscopic figures with the combination TILs low/high and TRGs? It would definitely add up to the paper`s value.
8) Line 209: The statement “provokes an inflammatory infiltrate of T cells.” is not shown. If you claim this, you have to show it with CD3+ staining.
9) Line 223-226: Statement “Our results might explain findings reported in these studies. The improved prognosis with lymph node regression might be at least partially explained by the associated tumor infiltrating lymphocytes it provokes.” You do not have data on patients´ survival so it is difficult to claim that your results support these results. In addition, you also showed that TILs are associated with lymph node positivity with regression characteristics. So how would you explain this?
Discussion section:
Maybe you could comment on the papers:
1) “Prognostic factors of perioperative FLOT regimen in operable gastric and gastroesophageal junction tumors: real-life data (Turkish Oncology Group)” since they used hematological values and you used tumor slides
2) Effect of Neoadjuvant Chemotherapy on Tumor-Infiltrating Lymphocytes in Resectable Gastric Cancer: Analysis from a Western Academic Center by Elliott J. Yee et al that partially overlaps with your study
Minor comments
1) Title: Hyphen should be added between Tumor-infiltrating
1) Line 8: Shouldn`t the institution be: 1st Propaedeutic (“e” instead of “i”) Department of Surgery?
2) In case you decide to keep TILs table 5, Fequency 5 should go before 10, and not behind 40.
Author Response
Dear reviewer,
Thank you for your comments as they are much appreciated.Please find attached a word document with a point-by-point revision according to your feedback.
Sincerely yours

Round 2
Reviewer 2 Report
Comments and Suggestions for Authors
In my opinion, the manuscript has been thoroughly corrected and improved.
I would like to address the results section, again tables in particular- I am aware that SPSS gives you tables in the same way as Graphpad or any other statistical program gives tables, however, it is on the author to take out important data from the tables.
The majority of these tables are just proof that you did the correct statistical analysis. That is something that the author himself needs to know and maybe the reviewer but not the general audience.
Therefore, if the authors and Editor agree, I would remove:
1) Tables 1-4 and instead of them put one cumulative Table (currently table 16, with correction in TIL low and high numbers and frequency as there is, I guess, a copy-paste error)
2) Table 6 since the results of the TIL high and TIL low groups are described in the text and there is no need for the Table.
3) Table 7, as results are described in the text and the Table is not necessary.
4) Tables 9 and 10 are completely unnecessary since with them you just confirm that you have the correct sample size to obtain valid results. Therefore, I would remove them especially since you are showing in Table 11 that you had the correct (not too small) sample size.
Minor comments:
In Table 5 what is in the first line where Frequency is 6 and Percent 8?
Line 145: Please add "T" into "TIL" "Figure 1. IL high patient."
Author Response
Thank you for your comments.
We have modified the tables as you suggested
- We have addressed the tables as you suggested. Thank you for making these suggestions.
- Addressed as instructed
- Thank you for the comments. We have revised the manuscript as you suggested
- We have removed the tables as instructed
Minor comment addressed: Thank you for pointing out in the figure.